

# Prediction study of prognostic nutrition index on the quality of life of patients with cervical cancer undergoing radiotherapy and chemotherapy

Ying Chen[1,*], Bifen Huang[2,*], Jianqing Zheng[1] and Fangjie He[3,4]

[1] Department of Radiation Oncology, The Second Affiliated Hospital of Fujian Medical University, Quanzhou, Fujian, China
[2] Department of Obstetrics and Gynecology, Quanzhou Medical College People's Hospital Affiliated, Quanzhou, Fujian, China
[3] Department of Obstetrics and Gynecology, The First People's Hospital of Foshan, Foshan, Guangdong, China
[4] State Key Laboratory of Oncology in South China, Sun Yat-sen University Cancer Center, Guangzhou, Guangdong, China
* These authors contributed equally to this work.

Corresponding authors
Jianqing Zheng,
18060108268@189.cn
Fangjie He, dr_hefangjie@163.com

## ABSTRACT

**Objective:** To assess the prognostic nutritional index (PNI) and quality of life (QOL) of patients with cervical cancer (CC) who underwent radiotherapy and chemotherapy and to reveal the effect of PNI on QOL and its prognostic value.
**Methods:** A total of 138 CC patients who underwent radiotherapy and chemotherapy in the Second Affiliated Hospital of Fujian Medical University from January 2020 to October 2022 were selected as the study subjects *via* convenient sampling. According to the PNI cut-off value of 48.8, they were divided into a high-PNI group and a low-PNI group, and the quality of life of the two groups was compared. The Kaplan-Meier method was used to draw the survival curve, and the Log-Rank test was employed to compare the survival rates of the two groups.
**Results:** The scores of physical functioning and overall QOL in the high-PNI group were significantly higher than those in the low-PNI group ($P < 0.05$). The scores of fatigue, nausea and vomiting, pain and diarrhea were higher than those in the low-PNI group, and the difference was statistically significant ($P < 0.05$).
The objective response rates were 96.77% and 81.25% in the high-PNI group and the low-PNI group, respectively, and the difference was statistically significant ($P = 0.045$). The 1-year survival rates of patients with high PNI and low PNI were 92.55% and 72.56% in the high-PNI group and the low-PNI group, respectively; the difference in survival rates was statistically significant ($P = 0.006$).
**Conclusion:** The overall quality of life of CC patients with low PNI receiving radiotherapy and chemotherapy is lower than that of patients with high PNI. Low PNI reduces the tolerance to radiotherapy and chemotherapy and the objective response rate, which can be used as a prognostic indicator for cervical cancer patients.

## INTRODUCTION

According to the global cancer statistics in 2022, the incidence of cervical cancer (CC) ranks seventh among all female malignancies, and it is one of the most common gynecological malignancies, posing a serious threat to women's physical and mental health (*Siegel et al., 2022*). China is ranked second in the world in the incidence of cervical cancer, accounting for about 500,000 new cases and 250,000 deaths annually (*Feng et al., 2019*; *Xia et al., 2022*). Moreover, the incidence of CC increased significantly with age before the age of 59, and the period effect of incidence and mortality exhibited an overall upward trend (*Sun et al., 2022*). Concurrent chemoradiation (CCRT) is the standard and the most important therapy for patients with local advanced CC unsuitable for surgery (*Zhang & Gao, 2022*). In recent years, intensity-modulated radiotherapy (IMRT) has become the standard radiotherapy scheme for cervical cancer, which not only improves the local radiation dose of the tumor, but also reduces the radiation dose of normal tissues and reduces the radiotherapy-related toxic reactions (*Lei et al., 2019*; *Zhang et al., 2022*). However, it had been shown that 30–40% of cervical cancer patients suffered from gastrointestinal side effects related to radiotherapy, resulting in severe diarrhea, insufficient food intake, and eventually malnutrition, which seriously affected the quality of life and prognosis of patients (*Aredes, Garcez & Chaves, 2018*). For cervical cancer patients receiving radiotherapy and chemotherapy, accurate assessment of quality of life and prognosis is conducive to the development of targeted treatment and nursing programs.

It has been reported that the prevalence of malnutrition was 17.7% before CCRT and 47.1% at the end of CCRT (*Argefa & Roets, 2022*). Malnutrition is associated with poor prognosis of CC and early detection of malnutrition and nutritional interventions could improve the outcome for cervical cancer patients. Patients who were malnourished had an increased risk of mortality (Hazard Ratio (HR): 3.12, 95% CI [1.23–7.86]) as compared to those who were well nourished for cervical cancer patients. There are many practical scales that can be used for early nutritional evaluation, such as NRS 2002, PG-SGA Tool, *etc.*, but these scales are subjective (*Ciebiera et al., 2021*). The prognostic nutritional index (PNI) is an index proposed by Japanese scholars to assess nutritional status, which is calculated from plasma albumin and peripheral blood lymphocyte count (*Demirelli et al., 2021*). In recent years, the application of PNI has been gradually extended to the field of oncology and it is considered to be used to assess the prognosis of cancer patients (*Dai et al., 2019*). Some studies have shown that nutrition intervention based on PNI can maintain the weight of patients, improve the tolerance of radiotherapy and chemotherapy, and ultimately improve the long-term efficacy (*Ge et al., 2021*; *Fanetti et al., 2021*). However, there is no consensus on the prognostic value of PNI in patients with cervical cancer undergoing radiotherapy and chemotherapy (*Haraga et al., 2016*). The PNI may be significantly associated with clinical complete response to chemoradiation in patients with cervical cancer undergoing radiotherapy and chemotherapy (*Gangopadhyay, 2020*), and may be a predictor of survival in patients with recurrent cervical cancer (*Ida et al., 2018*).

Therefore, it is useful to determine whether PNI has predictive value for the quality of life and prognosis of patients with cervical cancer.

To address this, the relationship between PNI and quality of life (QOL) was explored in clinical settings. Our study is aimed to assess the impact of PNI status on quality of life and treatment outcomes of cervical cancer patients. Over the years, our department has placed emphasis on the assessment and prediction of survival quality of cancer patients and has trained full-time quality-of-life study nurses to provide care and quality-of-life guidance to cancer patients. In this study, we analyzed the predictive effect of PNI on the survival quality of cervical cancer patients receiving radiotherapy and chemotherapy through a survey and data collection by a full-time research nurse. This will enable oncologists and other health care providers to use the PNI to reassess the nutritional status of CC patients and develop nutritional care plans to provide high-quality care and improve QOL for CC patients.

## PATIENTS AND METHODS

### Study design

A prospective trial was conducted in a single medical center in women patients with cervical cancer undergoing radiotherapy and chemotherapy between January 2020 and October 2022. The main purpose of the study is to explore the effect of PNI on the quality of life and prognosis of cervical cancer. The present study was approved by the Ethics Committee of the Second Affiliated Hospital of Fujian Medical University (Fujian, China, Approved NO: 2019-161) and was conducted according to the Declaration of Helsinki. The registration number of this trial is ChiCTR2300070480. Patients provided informed written consent at the time of data collection. A convenience sampling method was used to select patients, and study procedures were carried out in accordance with relevant guidelines and regulations (*Wild, Kyröläinen & Kuperman, 2022*). The reason for using convenient sampling was that the subjects come from multiple diagnosis and treatment groups. In order to ensure a sufficient sample size, we have adopted a convenient sampling method and established inclusion and exclusion criteria as follows.

### Patients

A total of 142 cases of cervical cancer treated with radiotherapy and chemotherapy in the Second Affiliated Hospital of Fujian Medical University from January 2020 to October 2022 were selected as the study population. Eligible patients were: (1) at least 18 years of age; (2) patients with pathologically confirmed cervical cancer; (3) patients with stage I–II cancer who required postoperative radiotherapy due to the existence of middle- and high-risk factors; or patients with stage II–III, inoperable cancer who required radical concurrent chemoradiation; or patients with stage IV cervical cancer requiring palliative radiotherapy for reasons such as combined vaginal bleeding and painful bone metastases; (3) patients with normal Chinese reading and writing ability and with normal mental state can cooperate to complete the research; (4) patients who are willing to adhere to regular follow-up visits; (5) patients who have consented and signed the informed consent form. Exclusion criteria were (1) patients with severe hepatic and renal insufficiency; (2) patients
with a combination of other tumors; (3) patients with severe psychiatric comorbidities (*e.g.*, schizophrenia, Korsakoff's syndrome and severe dementia). This study is in accordance with the Declaration of Helsinki and was approved by the Ethics Committee of the Second Affiliated Hospital of Fujian Medical University.

## Procedure

According to the clinical research rules, two senior attending physicians and two specialist nurses were trained to be dedicated research staff, who were responsible for health education, care and treatment guidance, quality of life surveys and data collection and collation for cervical cancer patients. The survival outcomes were followed up and registered by the treating physician.

Eligible patients were invited to participate the present study by their treating physician. On-site self-report surveys or online self-report questionnaires was used to collect relevant data and the investigators will provide answers and help if needed. Before the survey, the surveyors would introduce the purpose and meaning of the questionnaire and the way to fill it out to the patients using unified instruction. If patients are unable to fill out the questionnaire themselves, the surveyor can write the questionnaire for them.
The questionnaires were distributed and collected on the spot, and the completeness of the questionnaires was checked again after collection, and invalid questionnaires were excluded. The quality-of-life questionnaires should be completed independently within 1 h. The data collection and collation process were carried out in the form of double-checked entry to ensure the accuracy and completeness of data entry. Flow chart of study design was shown in Fig. 1.

## Measures

Clinical and demographic characteristics were collected *via* self-report questionnaires and medical records, including age, educational status, pathological type, healthcare provider payment method, the international federation of gynecology and obstetrics (FIGO) stage, the eastern cooperative oncology group (ECOG) performance status, treatment mode, bone metastasis pain, treatment modality, *etc.*

PNI score was calculated as serum albumin (g/L) + 5 × total lymphocyte count (n/L) (*Hu et al., 2021*). The total lymphocyte count and serum albumin were collected by a dedicated research nurse within 1 week after hospitalization. Patients with a lower PNI showed a worse nutritional status. Typically, PNI ≥ 48.8 is defined as high PNI status and PNI < 48.8 is low PNI status according to literature criteria (*Wang, Zhao & He, 2021*). In our prospective study, eligible cervical cancer patients were grouped according to a PNI threshold of 48.8.

The quality of life was assessed using the European Organisation for Research and Treatment of Cancer (EORTC) quality of life questionnaire QLQ-C30 (EORTC QLQ-C30). (*Cheng et al., 2011*; *Jang et al., 2010*). The EORTC QLQ-C30 is a self-administered, cancer-specific questionnaire with multidimensional scales (*Aaronson et al., 1993*). The EORTC QLQ-C30 consisted of five functional domains including physical, role, emotional, cognitive, and social; three symptom domains of fatigue, nausea/vomiting, and

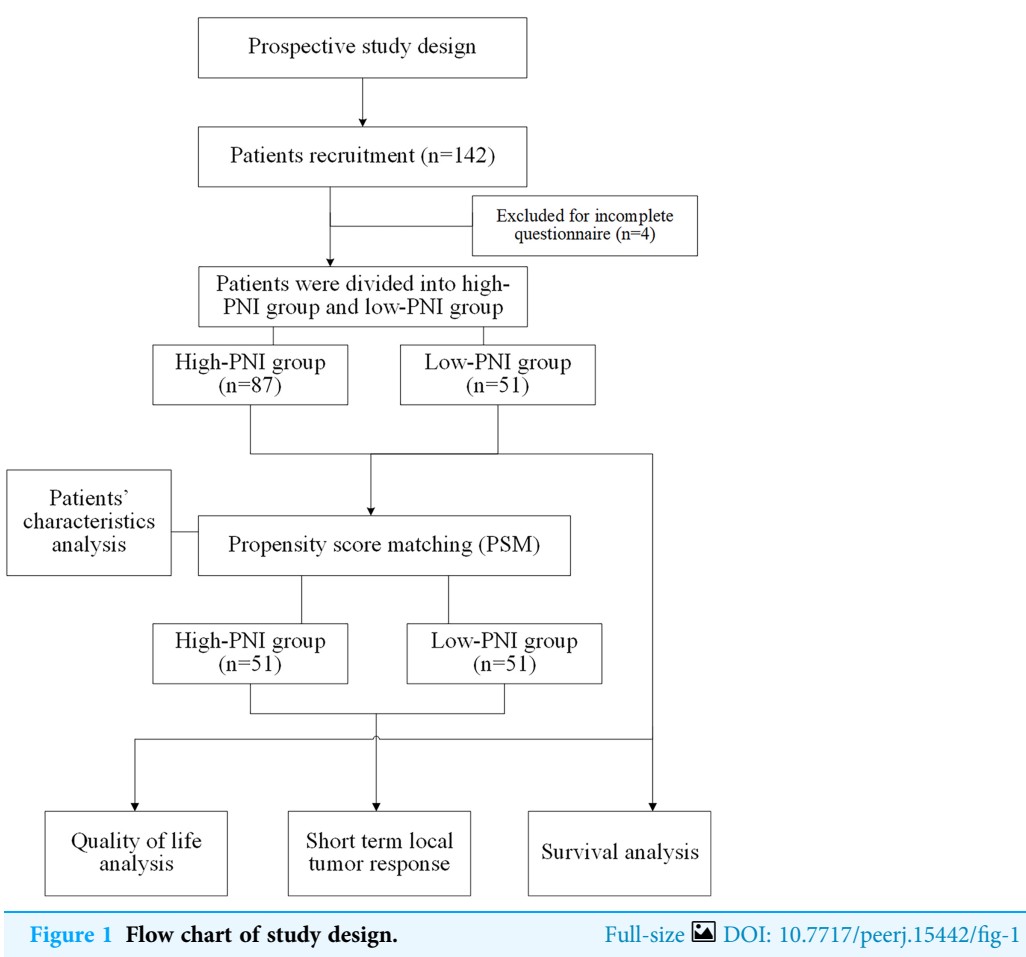

**Figure 1** Flow chart of study design.

pain; six individual-item symptom scores of dyspnea, sleep, appetite, constipation, diarrhea, and financial impact, and a global health domain to assess the overall health in the past week. All items are rated on a Likert scale. The EORTC QLQ-C30 is reliable and has been applied and validated in the Chinese cervical cancer population and some other sample of cancer (*Thapa et al., 2018*; *Bernardo, Li & Jimeno, 2018*). The reliability of EORTC QLQ-C30 was good (Cronbach's α = 0.77 to 0.88) (*Bernardo, Li & Jimeno, 2018*).

Short term local tumor response was evaluated by the response evaluation criteria in solid tumors (RECIST 1.1) (*Schwartz et al., 2016*). Complete response (CR) was defined as complete resolution of disease on magnetic resonance imaging (MRI), including reduction of the primary tumor and positive lymph nodes. Partial response (PR) was defined as the reduction of target lesions by more than 30%. Progressive disease (PD) is determined by progressive disease visible in imaging or gynecological examination. The objective response rates were calculated by the percentage of CR and PR among all treated patients, which was evaluated after administering the corresponding therapy for at least 4 weeks.

Overall survival (OS) was defined as the time from the date of treatment administered to the date of death due to any cause. In the absence of confirmation of death, survival time will be censored at the last date if the subject is known to be alive. Patients were followed-up every 3 months in the first year, every 4 months in the second year, 6-monthly

in the third year and annually after treatment. The follow-up work is completed by the attending physician.

## Statistical analysis

Standard descriptive analyses were performed to assess the clinical and demographic characteristics of sample. Only patients with complete data were included in the final statistical analysis. Descriptive statistics were applied to characterize the population (means and standard deviations (SD) for continuous variables and percentages for categorical variables). Differences in demographic and clinical variables were explored using t-tests, chi-square tests and ANOVA. Ranked ordinal data was analyzed *via* Wilcoxon rank sum test. Propensity score matching analysis (PSM) was applied to correct for background variables to achieve statistical comparability. Survival estimation was performed by the Kaplan-Meier method, and compared using the log-rank test. Uni- and multivariable Cox proportional-hazards model were applied to compare survival outcome between the two groups, where baseline variables were adjusted. All the statistical analyses were performed using R 4.2.0 software package *via* various R packages. A *p*-value of less than 0.05 was considered significant.

## RESULTS

### Patients' characteristics

A total of 142 questionnaires were distributed and 138 valid questionnaires were recovered, with a valid recovery rate of 97.18%. The completed time for each questionnaire was about (25.56 ± 5.32) mins. In 138 patients with completed data, the subjects were aged (58.45 ± 6.29) years, the youngest was 45 years old and the oldest was 75 years old. There were 110 (79.71%) patients with squamous squamous cell carcinoma, 21 (15.22%) patients with adenocarcinoma and 7 (5.07%) patients with adenosquamous carcinoma. There were 17 (12.32%) patients suffered cancer pain because of bone metastases.

According to the grouping threshold of PNI, 87 cases of cervical cancer were divided into the high-PNI group and 51 cases into the low-PNI group. Compared with the patients in the high-PNI group, the patients with cervical cancer in the low-PNI group had a higher age ($P$ = 0.008). More patients (56.86%) in the low-PNI group received concurrent radiotherapy and chemotherapy, while more patients (60.92%) in the high-PNI group received postoperative therapy. Patients in the high-PNI group have higher education level, of whom only 21.84% received primary or junior education. There were no significant differences regarding nationality, marriage, FIGO stage, ECOG, complication, comorbidity and pathological type. Demographic and medical characteristics are presented in Table 1.

To enable statistical comparability of important measures, propensity score matching analysis was carried out and the relevant results are presented in Table 2. After matching, the background variables were comparable.

**Table 1 Sociodemographic and clinical characteristics (N = 138).**

| Factors | Overall (*n* = 138) | High-PNI group (*n* = 87) | Low-PNI group (*n* = 51) | *P* |
|---|---|---|---|---|
| PNI (mean (SD)) | 49.87 (5.89) | 53.43 (3.53) | 43.80 (3.75) | <0.001 |
| Age | | | | |
| Median (IQR) | 58.00 (54.00, 63.00) | 58.00 (53.00, 62.00) | 61.00 (56.00, 64.00) | 0.008 |
| >60 years | 54 (39.13) | 27 (31.03) | 27 (52.94) | 0.018 |
| ≤60 years | 84 (60.87) | 60 (68.97) | 24 (47.06) | |
| Nationality | | | | |
| Han nationality | 108 (78.26) | 67 (77.01) | 41 (80.39) | 0.802 |
| Non-Han nationality | 30 (21.74) | 20 (22.99) | 10 (19.61) | |
| Marriage | | | | |
| Married | 105 (76.09) | 70 (80.46) | 35 (68.63) | 0.173 |
| Unmarried | 11 (7.97) | 7 (8.05) | 4 (7.84) | |
| Divorced | 22 (15.94) | 10 (11.49) | 12 (23.53) | |
| Education | | | | |
| Primary or junior education | 42 (30.43) | 19 (21.84) | 23 (45.10) | 0.013 |
| Senior or higher professional education | 60 (43.48) | 44 (50.57) | 16 (31.37) | |
| University or postgraduate education | 36 (26.09) | 24 (27.59) | 12 (23.53) | |
| Payment | | | | |
| Staff medical insurance | 38 (27.54) | 29 (33.33) | 9 (17.65) | 0.037 |
| Residents, new rural cooperative medical insurance | 46 (33.33) | 31 (35.63) | 15 (29.41) | |
| Other payments | 30 (21.74) | 17 (19.54) | 13 (25.49) | |
| Self-pay | 24 (17.39) | 10 (11.49) | 14 (27.45) | |
| Clinical stage (FIGO stage) | | | | |
| Stage I | 19 (13.77) | 13 (14.94) | 6 (11.76) | 0.663 |
| Stage II | 42 (30.43) | 29 (33.33) | 13 (25.49) | |
| Stage III | 48 (34.78) | 28 (32.18) | 20 (39.22) | |
| Stage IV | 29 (21.01) | 17 (19.54) | 12 (23.53) | |
| Treatment | | | | |
| Postoperative radiotherapy | 39 (28.26) | 30 (34.48) | 9 (17.65) | 0.011 |
| Postoperative radiotherapy and chemotherapy | 30 (21.74) | 23 (26.44) | 7 (13.73) | |
| Concurrent chemoradiotherapy | 57 (41.30) | 28 (32.18) | 29 (56.86) | |
| Palliative radiotherapy | 12 (8.70) | 6 (6.90) | 6 (11.76) | |
| ECOG | | | | |
| ECOG ≤2 | 98 (71.01) | 66 (75.86) | 32 (62.75) | 0.148 |
| ECOG >2 | 40 (28.99) | 21 (24.14) | 19 (37.25) | |
| Complication | | | | |
| No | 103 (74.64) | 66 (75.86) | 37 (72.55) | 0.819 |
| Yes | 35 (25.36) | 21 (24.14) | 14 (27.45) | |
| Comorbidity | | | | |
| No | 103 (74.64) | 65 (74.71) | 38 (74.51) | 0.999 |
| Yes | 35 (25.36) | 22 (25.29) | 13 (25.49) | |
| Pathological type | | | | |

(Continued)

| Table 1 (continued) | | | | |
|---|---|---|---|---|
| Factors | Overall (*n* = 138) | High-PNI group (*n* = 87) | Low-PNI group (*n* = 51) | *P* |
| Squamous cell carcinoma | 110 (79.71) | 73 (83.91) | 37 (72.55) | 0.250 |
| Adenocarcinoma | 21 (15.22) | 11 (12.64) | 10 (19.61) | |
| Adenosquamous carcinoma | 7 (5.07) | 3 (3.45) | 4 (7.84) | |
| BonePain | | | | |
| No | 121 (87.68) | 77 (88.51) | 44 (86.27) | 0.907 |
| Yes | 17 (12.32) | 10 (11.49) | 7 (13.73) | |

## Quality of life analysis in different PNI groups

The scores of physical functioning and overall quality of life in the high-PNI group were significantly higher than those in the low-PNI group both in pre-PSM population and post-PSM population ($P < 0.05$). The scores of social functioning, fatigue, nausea and vomiting, pain, sleeplessness and diarrhea were higher than those in the low-PNI group in pre-PSM population, and the difference was statistically significant ($P < 0.05$). However, the scores of fatigue, nausea and vomiting, pain and diarrhea were higher than those in the low-PNI group in post-PSM population, and the difference was statistically significant ($P < 0.05$). The results were shown in Figs. 2, 3, Tables 3 and 4.

## Short term local tumor response

A total of 28 patients with cervical cancer in the high-PNI group received radical concurrent chemoradiotherapy, and three patients received palliative radiotherapy at the primary site (to stop bleeding and improve local control), while 29 patients in the low-PNI group received radical concurrent chemoradiotherapy, and three patients received palliative radiotherapy at the primary site see Table 1. Based on the modified RECIST 1.1 criteria, the objective response rates were 96.77% and 81.25%, respectively, and the difference was statistically significant ($P = 0.045$) see Table 5. Further analysis showed that three patients in the low-PNI group had delayed radiotherapy due to grade 3/4 radioactive enteritis, and two patients automatically abandoned treatment because of radiotherapy intolerance.

## Survival analysis in different PNI groups

The median follow-up period was 15.2 months. In the non-PSM population, the 1-year survival rates of patients with high PNI and low PNI were 93.93% (95% CI [88.33–99.88%]) and 72.56% (95% CI [57.05–92.30%]), respectively; the difference in survival rates was statistically significant ($P < 0.001$). In the PSM population, the 1-year survival rates of patients with high PNI and low PNI were 92.55% (95% CI [84.78–96.54%]) and 72.56% (95% CI [57.05–92.30%]), respectively; the difference in survival rates was statistically significant ($P = 0.006$). The Kaplan-Meier curves of the two groups are shown in Figs. 4 and 5.

Survival analysis results of univariable Cox proportional-hazards model were show in Table 6. Survival analysis results of multivariable Cox proportional-hazards model were

**Table 2 Sociodemographic and clinical characteristics of propensity score matched population (N = 102).**

| Factors | High-PNI group (n = 51) | Low-PNI group (n = 51) | P |
|---|---|---|---|
| PNI (mean (SD)) | 53.27 (3.31) | 43.80 (3.75) | <0.001 |
| Age | | | |
| Median (IQR) | 56.00 (44.00, 62.00) | 56.00 (44.00, 63.00) | 0.385 |
| >60 years | 22 (43.14) | 27 (52.94) | 0.428 |
| ≤60 years | 29 (56.86) | 24 (47.06) | |
| Nationality | | | |
| Han-nationality | 37 (72.55) | 41 (80.39) | 0.484 |
| Non-Han-nationality | 14 (27.45) | 10 (19.61) | |
| Marriage | | | |
| Married | 39 (76.47) | 35 (68.63) | 0.674 |
| Unmarried | 3 (5.88) | 4 (7.84) | |
| Divorced | 9 (17.65) | 12 (23.53) | |
| Education | | | |
| Primary or junior education | 18 (35.29) | 23 (45.10) | 0.449 |
| Senior or higher professional education | 22 (43.14) | 16 (31.37) | |
| University or postgraduate education | 11 (21.57) | 12 (23.53) | |
| Payment | | | |
| Staff medical insurance | 10 (19.61) | 9 (17.65) | 0.83 |
| Residents, new rural cooperative medical insurance | 17 (33.33) | 15 (29.41) | |
| Other payments | 14 (27.45) | 13 (25.49) | |
| Self-pay | 10 (19.61) | 14 (27.45) | |
| Clinical stage (FIGO stage) | | | |
| Stage I | 8 (15.69) | 6 (11.76) | 0.87 |
| Stage II | 15 (29.41) | 13 (25.49) | |
| Stage III | 18 (35.29) | 20 (39.22) | |
| Stage IV | 10 (19.61) | 12 (23.53) | |
| Treatment | | | |
| Postoperative radiotherapy | 12 (23.53) | 9 (17.65) | 0.678 |
| Postoperative radiotherapy and chemotherapy | 10 (19.61) | 7 (13.73) | |
| Concurrent chemoradiotherapy | 24 (47.06) | 29 (56.86) | |
| Palliative radiotherapy | 5 (9.80) | 6 (11.76) | |
| ECOG | | | |
| ECOG ≤2 | 33 (64.71) | 32 (62.75) | 1 |
| ECOG >2 | 18 (35.29) | 19 (37.25) | |
| Complication | | | |
| No | 37 (72.55) | 37 (72.55) | 1 |
| Yes | 14 (27.45) | 14 (27.45) | |
| Comorbidity | | | |
| No | 40 (78.43) | 38 (74.51) | 0.815 |
| Yes | 11 (21.57) | 13 (25.49) | |
| Pathological type | | | |

*(Continued)*

| Table 2 (continued) | | | |
|---|---|---|---|
| Factors | High-PNI group (*n* = 51) | Low-PNI group (*n* = 51) | *P* |
| Squamous cell carcinoma | 39 (76.47) | 37 (72.55) | 0.883 |
| Adenocarcinoma | 9 (17.65) | 10 (19.61) | |
| Adenosquamous carcinoma | 3 (5.88) | 4 (7.84) | |
| BonePain | | | |
| No | 45 (88.24) | 44 (86.27) | 1.000 |
| Yes | 6 (11.76) | 7 (13.73) | |

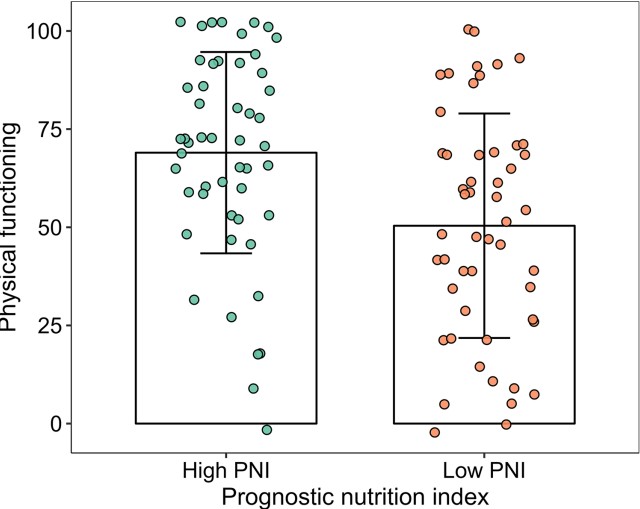

**Figure 2 Column and jitter diagram with error line for physical functioning.**

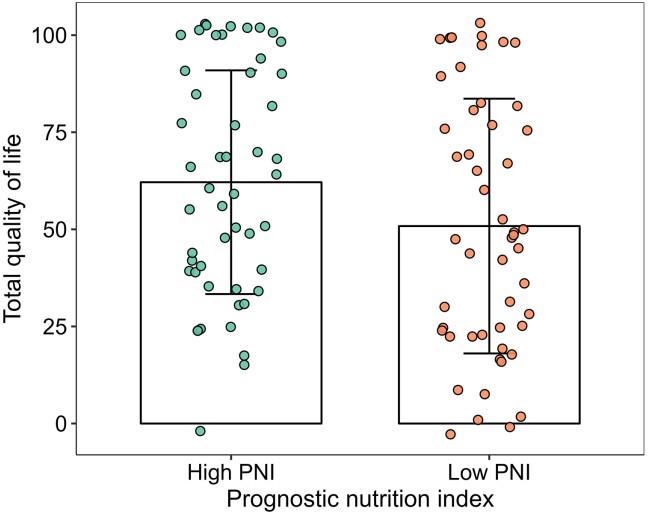

**Figure 3 Column and jitter diagram with error line for overall quality of life.**

**Table 3  Quality of life analysis in different PNI groups of the pre-PSM population.**

| Domain | High-PNI group (n = 87) | Low PNI group (n = 51) | t | P |
|---|---|---|---|---|
| Physical functioning | 73.85 ± 23.78 | 50.39 ± 28.59 | 5.186 | <0.001 |
| Role functioning | 68.36 ± 34.45 | 56.88 ± 33.16 | 1.915 | 0.058 |
| Emotional functioning | 68.45 ± 25.99 | 60.00 ± 30.26 | 1.733 | 0.085 |
| Cognitive functioning | 68.18 ± 31.93 | 57.18 ± 36.09 | 1.862 | 0.065 |
| Social functioning | 69.69 ± 31.06 | 56.88 ± 36.36 | 2.193 | 0.030 |
| Quality of life | 66.62 ± 28.74 | 50.84 ± 32.8 | 2.953 | 0.004 |
| Fatigue | 71.69 ± 29.74 | 50.75 ± 34.04 | 3.784 | <0.001 |
| Nausea and vomiting | 71.87 ± 33.22 | 50.08 ± 36.91 | 3.569 | <0.001 |
| Pain | 72.78 ± 31.34 | 48.02 ± 38.37 | 4.118 | <0.001 |
| Dyspnea | 68.57 ± 40.81 | 57.51 ± 43.79 | 1.496 | 0.137 |
| Sleeplessness | 69.71 ± 40.59 | 51.57 ± 40.78 | 2.53 | 0.013 |
| Appetite loss | 64.72 ± 42.68 | 56.84 ± 43.90 | 1.036 | 0.302 |
| Constipation | 71.62 ± 39.59 | 58.14 ± 44.68 | 1.841 | 0.068 |
| Diarrhea | 74.69 ± 40.69 | 48.39 ± 43.91 | 3.559 | 0.001 |
| Financial difficulties | 65.44 ± 41.81 | 58.75 ± 44.06 | 0.89 | 0.375 |

**Table 4  Quality of life analysis in different PNI groups of the post-PSM population.**

| Domain | High-PNI group (n = 51) | Low PNI group (n = 51) | t | P |
|---|---|---|---|---|
| Physical functioning | 69.00 ± 25.64 | 50.39 ± 28.59 | 3.461 | 0.001 |
| Role functioning | 64.00 ± 34.36 | 56.88 ± 33.16 | 1.064 | 0.290 |
| Emotional functioning | 70.06 ± 24.36 | 60.00 ± 30.26 | 1.849 | 0.067 |
| Cognitive functioning | 60.14 ± 33.75 | 57.18 ± 36.09 | 0.428 | 0.670 |
| Social functioning | 65.29 ± 34.43 | 56.88 ± 36.36 | 1.200 | 0.233 |
| Quality of life | 64.24 ± 28.80 | 50.84 ± 32.80 | 2.192 | 0.031 |
| Fatigue | 71.51 ± 28.71 | 50.75 ± 34.04 | 3.330 | 0.001 |
| Nausea and vomiting | 68.04 ± 31.53 | 50.08 ± 36.91 | 2.642 | 0.010 |
| Pain | 70.27 ± 31.49 | 48.02 ± 38.37 | 3.202 | 0.002 |
| Dyspnea | 66.65 ± 41.14 | 57.51 ± 43.79 | 1.086 | 0.280 |
| Sleeplessness | 63.37 ± 40.74 | 51.57 ± 40.78 | 1.462 | 0.147 |
| Appetite loss | 61.39 ± 42.93 | 56.84 ± 43.90 | 0.529 | 0.598 |
| Constipation | 65.31 ± 41.66 | 58.14 ± 44.68 | 0.839 | 0.404 |
| Diarrhea | 71.88 ± 42.38 | 48.39 ± 43.91 | 2.749 | 0.007 |
| Financial difficulties | 59.39 ± 43.42 | 58.75 ± 44.06 | 0.075 | 0.941 |

**Table 5 Analysis of short-term local tumor response in different PNI groups.**

| Groups | n | CR | PR | SD | ORR (%) | z* | P |
|---|---|---|---|---|---|---|---|
| High-PNI group | 31 | 22 (70.97) | 8 (25.81) | 1 (3.23) | 96.77 | 2.004 | 0.045 |
| Low PNI group | 32 | 16 (50) | 10 (31.25) | 6 (18.75) | 81.25 | | |

**Note:**
* Wilcoxon rank sum test.

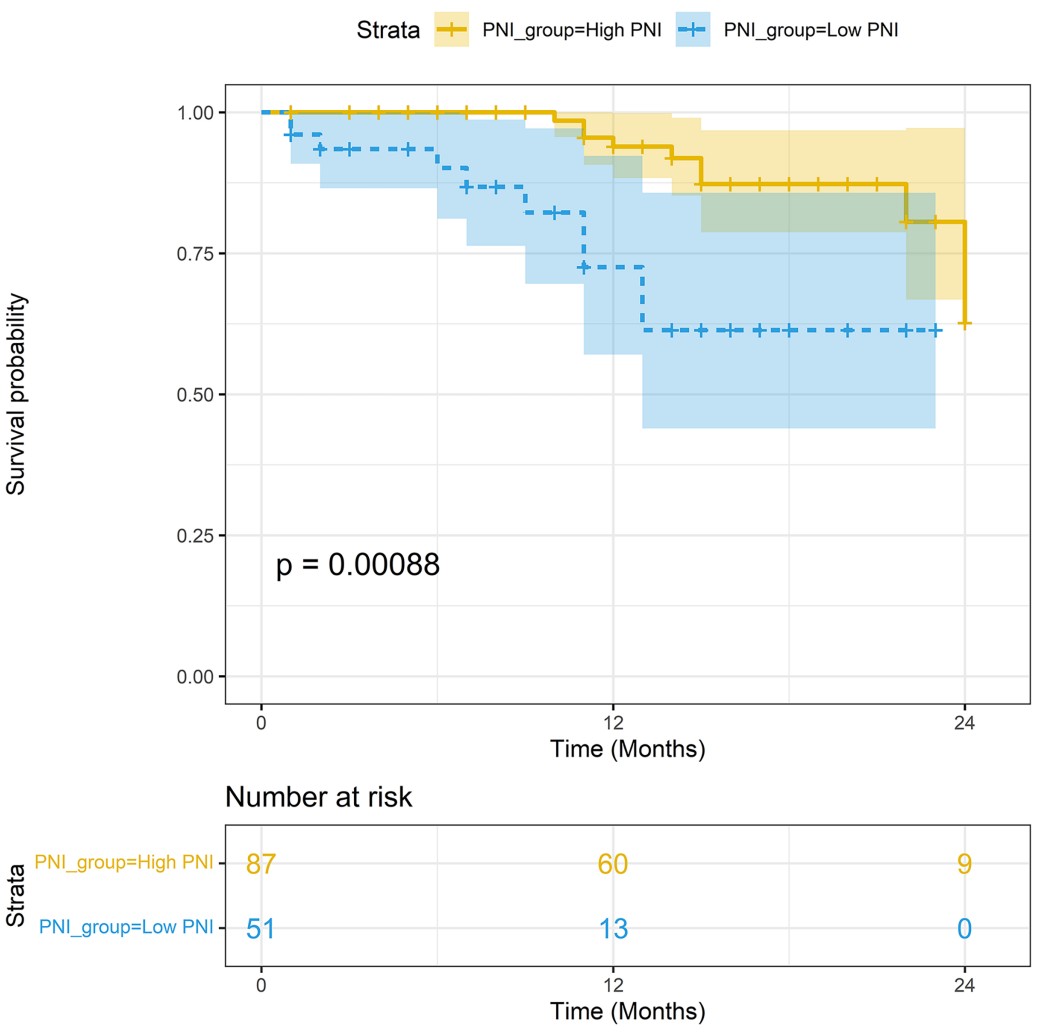

**Figure 4 Survival analysis between the high-PNI group and low-PNI group in non-PSM population.**

show in Table 7 and Fig. 6. Survival was worse in the low-PNI group than in the high-PNI group (adjusted HR: 3.679, 95% CI [1.125–12.029]), which showed that PNI was an important independent prognostic factor for cervical cancer.

A multivariate correlation analysis was used to analyze the correlation between PNI and other baseline variables to explore their interactions. The results are shown in Fig. 7. The results have shown that some variables are significantly correlated, but most of the correlation coefficients were below 0.3.

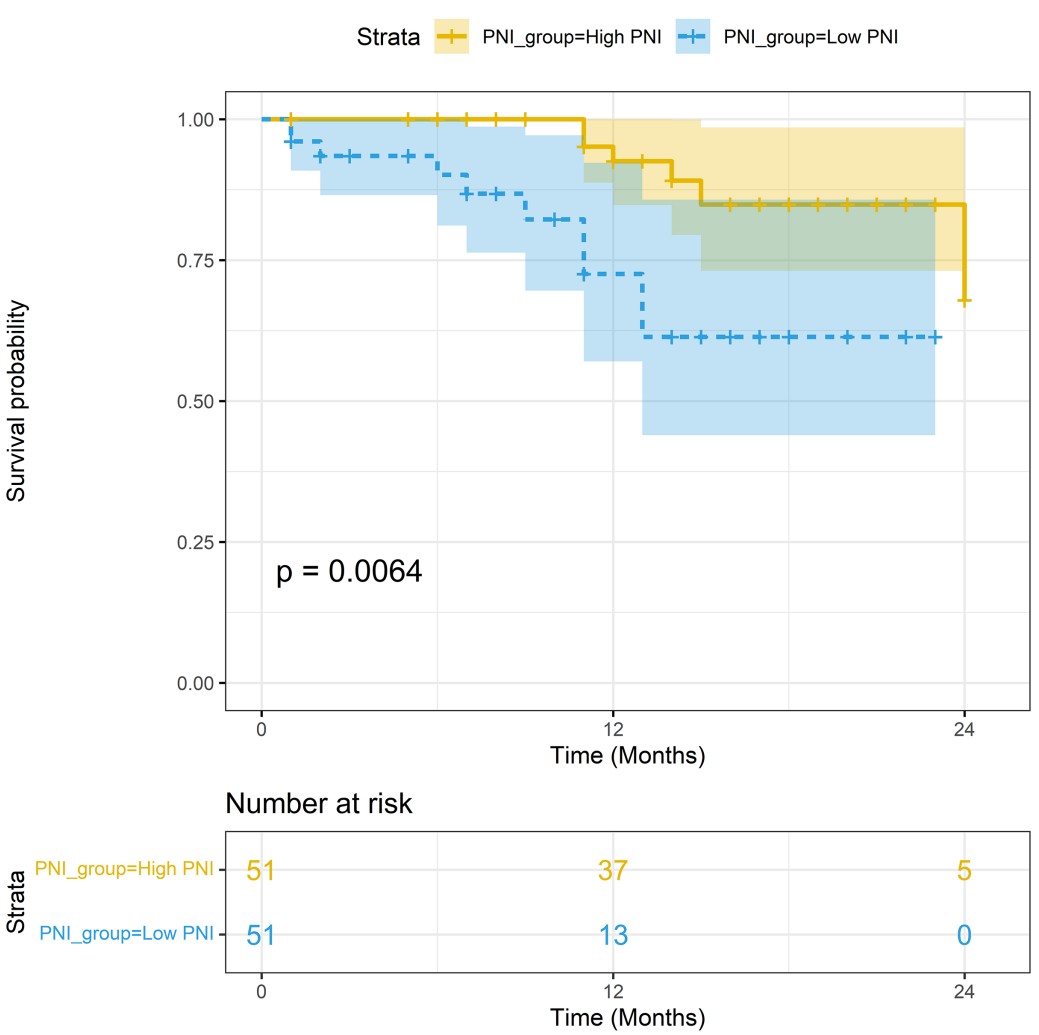

**Figure 5 Survival analysis between the high-PNI group and low-PNI group in PSM population.**

## DISCUSSION

Cancer patients are prone to nutritional problems, and the incidence of moderate and severe malnutrition in hospitalized cancer patients is as high as 60–80% (*Chen, Jiang & He, 2021*). The deaths due to malnutrition account for about 20–22%, which is the main cause of death for patients with cancer (*Schuetz et al., 2021*). Cancer-related malnutrition has become the focus of research aimed at improving the treatment and prognosis of cancer patients (*Schuetz et al., 2021*). There is no universal agreement about the best diagnostic criteria, the definition of malnutrition and how to identify the patients who would benefit from nutritional intervention (*Schuetz et al., 2021*; *Elia, 2017*). Consequently, nutrition assessment is one of the most important contents of comprehensive treatment of cancer (*Seo et al., 2022*). The most commonly used nutritional screening tools are various scales, diversely developed for detection of malnutrition, management of malnutrition, and prediction of clinical outcomes or health care usage, among which nutritional risk

**Table 6 Univariable Cox proportional-hazards model analysis.**

| Varname | Level | β | HR with 95% CI | Z | P |
|---|---|---|---|---|---|
| PNI | Low PNI | 1.470 | 4.351 [1.701–11.126] | 3.069 | 0.002 |
| Age | ≤60 years | 0.168 | 1.183 [0.453–3.087] | 0.344 | 0.731 |
| Nationality | Non-Han-nationality | 0.146 | 1.157 [0.413–3.244] | 0.277 | 0.782 |
| Marriage | Unmarried | 0.477 | 1.611 [0.365–7.115] | 0.629 | 0.529 |
| | Divorced | −0.483 | 0.617 [0.142–2.686] | −0.643 | 0.520 |
| Education | Senior or higher professional education | −1.070 | 0.343 [0.132–0.891] | −2.196 | 0.028 |
| | University or postgraduate education | −1.412 | 0.244 [0.054–1.101] | −1.835 | 0.067 |
| Payment | Residents, new rural cooperative medical insurance | 0.409 | 1.505 [0.424–5.340] | 0.632 | 0.527 |
| | Other payments | 1.040 | 2.829 [0.804–9.959] | 1.619 | 0.105 |
| | Self-pay | 0.654 | 1.923 [0.428–8.638] | 0.853 | 0.394 |
| Clinical stage | Stage II | −0.964 | 0.381 [0.077–1.895] | −1.178 | 0.239 |
| | Stage III | −0.868 | 0.420 [0.085–2.083] | −1.062 | 0.288 |
| | Stage IV | 0.929 | 2.532 [0.705–9.096] | 1.424 | 0.154 |
| Treatment | Postoperative radiotherapy and chemotherapy | −1.375 | 0.253 [0.030–2.112] | −1.270 | 0.204 |
| | Concurrent chemoradiotherap | 0.347 | 1.414 [0.501–3.993] | 0.654 | 0.513 |
| | Palliative radiotherapy | 1.203 | 3.331 [0.911–12.178] | 1.819 | 0.069 |
| ECOG | ECOGECOG >2 | 0.732 | 2.079 [0.845–5.115] | 1.593 | 0.111 |
| Complication | Yes | 0.420 | 1.522 [0.584–3.970] | 0.859 | 0.390 |
| Comorbidity | Yes | −0.465 | 0.628 [0.184–2.149] | −0.741 | 0.459 |
| Pathological type | Adenocarcinoma | 0.259 | 1.296 [0.429–3.919] | 0.459 | 0.646 |
| | Adenosquamous carcinoma | 0.564 | 1.758 [0.227–13.614] | 0.540 | 0.589 |

**Table 7 Multivariable Cox proportional-hazards model analysis.**

| Varname | VarLevel | β | HR with 95% CI | Z | P |
|---|---|---|---|---|---|
| PNI | Low PNI | 1.303 | 3.679 [1.125–12.029] | 2.155 | 0.031 |
| Education | Senior or higher professional education | −0.427 | 0.652 [0.196–2.167] | −0.697 | 0.486 |
| | University or postgraduate education | −0.965 | 0.381 [0.067–2.171] | −1.087 | 0.277 |
| Payment | Residents, new rural cooperative medical insurance | −0.011 | 0.989 [0.218–4.482] | −0.014 | 0.988 |
| | Other payments | 0.649 | 1.914 [0.383–9.578] | 0.791 | 0.429 |
| | Self-pay | 0.368 | 1.445 [0.290–7.195] | 0.450 | 0.653 |
| Clinical stage | Stage II | −0.993 | 0.370 [0.065–2.100] | −1.122 | 0.262 |
| | Stage III | −1.137 | 0.321 [0.055–1.863] | −1.267 | 0.205 |
| | Stage IV | 0.789 | 2.202 [0.495–9.793] | 1.036 | 0.300 |
| Treatment | Postoperative radiotherapy and chemotherapy | −0.938 | 0.391 [0.044–3.459] | −0.844 | 0.399 |
| | Concurrent chemoradiotherap | −0.028 | 0.972 [0.229–4.128] | −0.038 | 0.970 |
| | Palliative radiotherapy | 0.230 | 1.258 [0.119–13.249] | 0.191 | 0.848 |
| ECOG | ECOG >2 | −0.395 | 0.674 [0.141–3.221] | −0.495 | 0.621 |

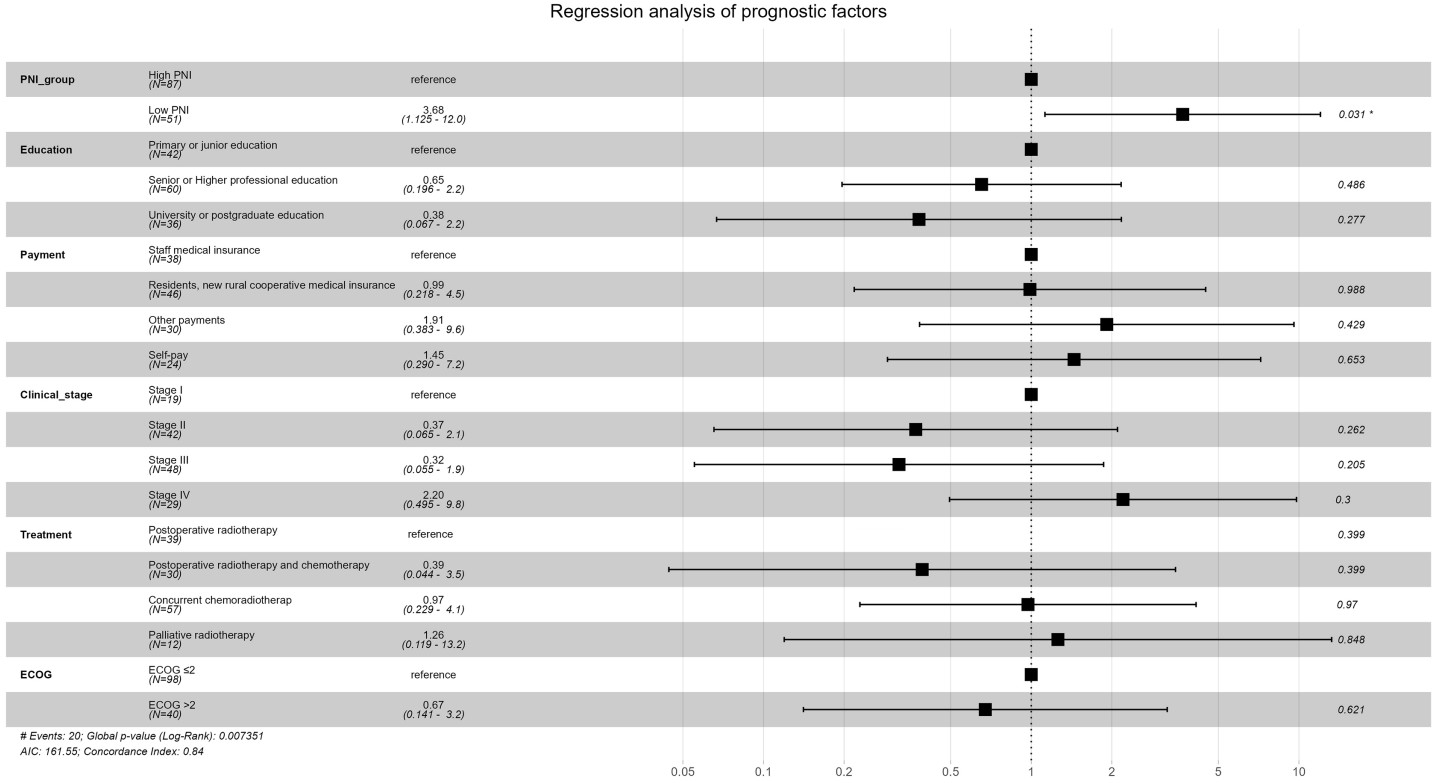

**Figure 6  Forest plot for multivariable Cox proportional-hazards model in the non-PSM population.**

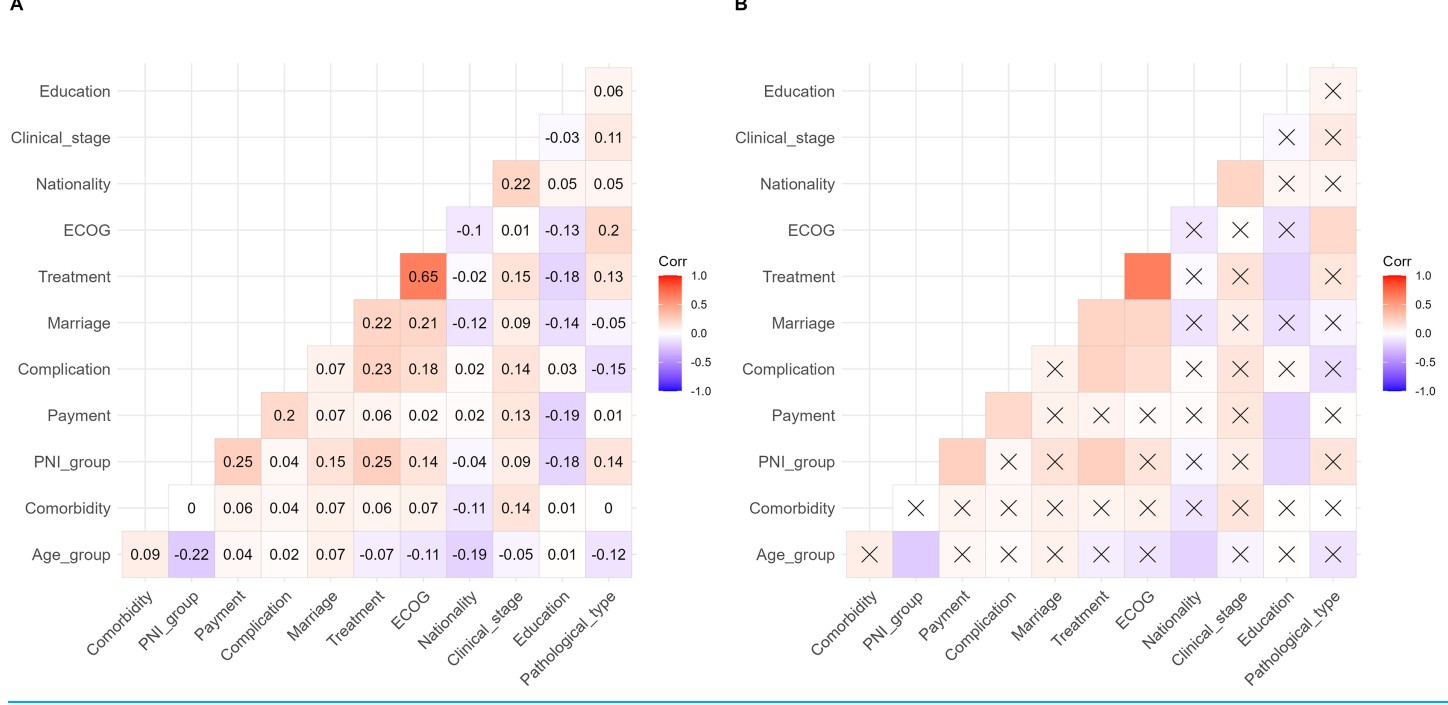

**Figure 7  Correlation matrix diagram of multivariate correlation analysis.**

screening 2002 (NRS2002), patient-generated subjective nutrition assessment (PG-SGA) and malnutrition universal screening tool (MUST) are the most commonly used in nutritional assessment for patients with cancer (*Elia, 2017*; *Miller et al., 2018*). However, these evaluation tools have limitations, which are relatively cumbersome for nutritional assessment and could easily increase the workload of nursing staff in clinical application.

PNI, also called Onodera's index, is calculated by selecting plasma albumin and total count of peripheral blood lymphocytes, which is a simple, convenient and practical nutritional index without complicated evaluation (*Wang, Zhao & He, 2021*). Albumin is the main protein component in the total serum protein of normal human body, and its decline can be caused by long-term insufficient protein intake or increased consumption of patients caused by malnutrition and chronic consumptive diseases, especially for cancer patients. Lymphocytes are a family of white blood cells found in the blood and lymphoid tissues that determines the specificity of immune responses to foreign organisms, which mediate the body's cellular immunity, humoral immunity and killing effect on tumor cells and virus-infected cells (*Van Der Leun, Thommen & Schumacher, 2020*). Accordingly, PNI can reflect the general condition of cancer patients including nutritional status, physical condition, protein turnover, and immune competence (*Demirelli et al., 2021*). PNI has been widely used to evaluate the nutritional status of esophageal cancer and head and neck cancer, and has been proved to reflect the prognosis of these cancer patients (*Ge et al., 2021*; *Fanetti et al., 2021*). However, the prognostic values of PNI in cervical cancer patients are still unknown, and data for the association between PNI and QOL are scanty, this study aims to investigate the association between PNI and survival quality for this malignancy.

First of all, we divided the subjects according to the PNI cut-off value of 48.8, that is, the high-PNI group and the low-PNI group. Currently, there is no unified standard for the cut-off value of PNI. Onodera et al. proposed that PNI > 50 represents the normal nutritional status, while PNI < 50 indicates different degrees of malnutrition (*Onodera, Goseki & Kosaki, 1984*); however, *Lee et al. (2017)* proposed that the PNI cut-off value of 49.5 for pancreatic cancer patients has better sensitivity and specificity. At present, the PNI cut-off value of 48.8 has become the most recognized grouping standard (*Fanetti et al., 2021*; *Migita et al., 2013*). In this study, with the PNI cut-off value of 48.8, 36.96% (51/138) of cervical cancers were screened and divided into low-PNI group, indicating that malnutrition in patients with cervical cancers is a serious problem. Second, we found that more patients with young age, more patients received radical surgery + adjuvant radiotherapy, and more patients with higher education level in the high-PNI group. We speculated that these young patients have higher educational level and are easy to detect their disease status early, so they are more likely to get early surgery. Third, we explored the differences of QOL between the different PNI groups. Taking into account differences in age, treatment mode, and education level, a propensity matching analysis was applied to correct the statistical heterogeneity between the two groups. And then statistical differences of scores of physical functioning and overall quality of life in the high-PNI group were detected in the low-PNI group. Moreover, patients in the low-PNI group were more likely to have treatment-related complications such as fatigue, nausea

and vomiting, pain and diarrhea. Fourth, we took the patients with cervical cancer who have received radical concurrent chemoradiotherapy with comparable primary tumor lesions as the research object to explore the objective response rate of tumor treatment under different PNI status. We found that the ORR of patients with low PNI was lower compared with patients with high PNI, and further found that the main reason for local treatment failure of patients was related to radiotherapy intolerance and radioactive enteritis. Finally, we compared the difference of OS under different PNI status. We found that patients with low PNI had worse prognosis.

As far as we know, there is no consensus on whether PNI is related to the QOL of cervical cancer, because high-quality research of this topic is still quite lacking, however, the research on QOL of cervical cancer has been the focus of gynecological oncology nurses (*Thapa et al., 2018*; *Medina-Contreras et al., 2020*). Our study shows that PNI is an important factor affecting the QOL of cervical cancer, which suggests that nutritional intervention could improve the QOL of patients with low PNI. In the different items of QLQ-C30 scale, the scores of fatigue, nausea and vomiting, pain and diarrhea in the high-PNI group were significantly higher than those in the low-PNI group, indicating that these symptoms in the high-PNI patients are less serious than those in the low-PNI group. Therefore, attention should be paid to strengthening the intervention of various symptoms of patients with low PNI in the daily clinical nursing work, so as to improve the quality of life of patients with low PNI. Meanwhile, the scores of physical function and overall quality of life in the high-PNI group were higher than those in the low-PNI group, suggesting that the nutritional status of patients may affect physical function and overall status of patients. Malnourished cancer patients are more likely to have abnormal physical function and bad mood than those with normal nutrition, which may be related to the feeling of weakness and poor mental state caused by malnutrition (*Zhang et al., 2021*).

Several studies have revealed that PNI was associated with survival in cervical cancer (*Haraga et al., 2016*; *Ida et al., 2018*). Malnutrition is considered to be an independent factor in the poor prognosis of cervical cancer (*Zhang et al., 2021*). In a similar study of cervical-cancer patients receiving CCRT, progression-free survival (PFS) and OS of patients with lower PNI were significantly shorter than those of patients with higher PNI with the PNI cutoff-value of 48.55 (*Haraga et al., 2016*). Multivariate analysis also found that low PNI was an independent prognostic factor for PFS and OS in patients receiving CCRT. Therefore, low PNI could predict poor prognosis in cancer patients (*Haraga et al., 2016*). *Ida et al. (2018)* had found that the OS of cervical cancer patients with low PNI was significantly worse than that of patients with PNI, and low PNI could reflect the nutritional decline of patients with recurrent cervical cancer. Our study suggests that PNI can also be used as a prognostic indicator for patients with CC undergoing radiotherapy and chemotherapy. Combined with the analysis results of QOL in this study, the overall QOL of CC patients with low PNI was lower than that of patients with high PNI.

The advantage of this study is to investigate the QOL in view of the scarcity of PNI data in patients with CC, providing a basis for the study of the relationship between QOL and nutritional status of CC. In addition, we use propensity matching analysis to effectively solve the heterogeneity of patient background data in the different PNI groups, and

enhance the reliability and effectiveness of research conclusions. This study also has some limitations that need to be considered. First of all, the sample size in this study is small and due to the differences in the baseline status of patients in the different PNI groups, only 102 valid samples were included in the final analysis, which indicates that we need to continue to expand case recruitment to verify the reliability of the results. Second, the observation time was short, and different stages of CC patients were included, from which we cannot draw a ground conclusion, so the survival outcome needs to be carefully explained. Finally, patients-selection bias may exist because a convenience sampling method was used in our study. Although convenience sampling is a common strategy which is the easiest, least time-intensive, and least expensive to implement, patient-selection bias may be an important risk affecting the robustness of outcomes.

## CONCLUSION

In conclusion, the overall quality of life of cervical cancer patients with low PNI receiving radiotherapy and chemotherapy is lower than that of patients with high PNI. Low PNI reduces the tolerance to radiotherapy and chemotherapy and the objective response rate, which can be used as a prognostic indicator for cervical cancer patients.

## ACKNOWLEDGEMENTS

We acknowledge all the clinicians, nurses, and interviewees who agreed to participate and give their opinions in this study.

### Funding

This study was supported by the China Postdoctoral Science Foundation (Grant No: 2021M700779 to Fangjie He), the Medical Research Fund of Guangdong Province (Grant No: A2022044 to Fangjie He), the Fujian provincial health technology project (Youth Scientific Research Project, 2019-1-50 to Jianqing Zheng), the Education and Scientific Research Project for Young and Middle-Aged Teachers of Fujian Provincial Department of Education (Grant No: JAT190223 to Ying Chen) and the Nursery Fund Project of the Second Affiliated Hospital of Fujian Medical University (Grant No: 2021MP05 to Jianqing Zheng). The funders had no role in study design, data collection and analysis, decision to publish, or preparation of the manuscript.

### Grant Disclosures

The following grant information was disclosed by the authors:
China Postdoctoral Science Foundation: 2021M700779.
Medical Research Fund of Guangdong Province: A2022044.
Fujian Provincial Health Technology project: 2019-1-50.
Education and Scientific Research Project for Young and Middle-Aged Teachers of Fujian Provincial Department of Education: JAT190223.

Nursery Fund Project of the Second Affiliated Hospital of Fujian Medical University: 2021MP05.

## Competing Interests

The authors declare that they have no competing interests.

## Author Contributions

- Ying Chen performed the experiments, analyzed the data, prepared figures and/or tables, and approved the final draft.
- Bifen Huang performed the experiments, analyzed the data, prepared figures and/or tables, and approved the final draft.
- Jianqing Zheng conceived and designed the experiments, authored or reviewed drafts of the article, and approved the final draft.
- Fangjie He conceived and designed the experiments, authored or reviewed drafts of the article, and approved the final draft.

## Human Ethics

The following information was supplied relating to ethical approvals (*i.e.*, approving body and any reference numbers):

This study was approved by The Second Affiliated Hospital of Fujian Medical University (2019-161).

## Data Availability

The raw data is available in the Supplemental File.

## Supplemental Information

Supplemental information for this article can be found online at http://dx.doi.org/10.7717/peerj.15442#supplemental-information.

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
