# Peer review of "Prediction study of prognostic nutrition index on the quality of life of patients with cervical cancer undergoing radiotherapy and chemotherapy"

_PeerJ, doi:10.7717/peerj.15442_

## Round 0.1 · original submission · Major Revisions

Please revise the manuscript carefully following reviewers' suggestions.

Reviewer 1 has suggested that you cite specific references. You are welcome to add it/them if you believe they are relevant. However, you are not required to include these citations, and if you do not include them, this will not influence my decision.

Reviewer 1 ·

Basic reporting

The authors explore the link between prognostic nutrition index and quality of life, hypothesizing that nutritional intervention could improve the quality of life of patients suffering from cervical cancer. In general terms, the article is interesting and covers an important topic.
The language is clear and technically correct. The tables present the data in a clear and complete way, the well-structured figures make the text more interesting. The background is good to determine the knowledge of the topic.

Experimental design

The research question results well defined. The investigation has been conducted rigorously and methods indicate in a detailed way materials, statistical analyses and procedures used.

Validity of the findings

The authors critically analyze the results, highlighting the underlying principles and trends that unite part or all of the results. Conclusions are well stated and linked to original research question. For completeness, it would be interesting to describe the quality of life from a psychological point of view
I would suggest read and use the following references:
Marano G, Traversi G, Gesualdi A, Biffi A, Gaetani E, Sani G, Mazza M. Mental Health and Coaching Challenges Facing the COVID-19 Outbreak. Psychiatr Danub. 2021 Spring;33(1):124-126. PMID: 33857059.
Marano G, Traversi G, Mazza M. Web-mediated Counseling Relationship in Support of the New Sexuality and Affectivity During the COVID-19 Epidemic: A Continuum Between Desire and Fear. Arch Sex Behav. 2021 Apr;50(3):753-755. doi: 10.1007/s10508-020-01908-3.
Suvaal I, Kirchheiner K, Nout RA, Sturdza AE, Van Limbergen E, Lindegaard JC, Putter H, Jürgenliemk-Schulz IM, Chargari C, Tanderup K, Pötter R, Creutzberg CL, Ter Kuile MM. Vaginal changes, sexual functioning and distress of women with locally advanced cervical cancer treated in the EMBRACE vaginal morbidity substudy. Gynecol Oncol. 2023 Mar;170:123-132. doi: 10.1016/j.ygyno.2023.01.005.
Membrilla-Beltran L, Cardona D, Camara-Roca L, Aparicio-Mota A, Roman P, Rueda-Ruzafa L. Impact of Cervical Cancer on Quality of Life and Sexuality in Female Survivors. Int J Environ Res Public Health. 2023 Feb 20;20(4):3751. doi: 10.3390/ijerph20043751.

Reviewer 2 ·

Basic reporting

1.The manuscript is generally easy to understand. However, the logic manner should be improved. For example, the introduction section is suggested to separate into three paragraphs. The first paragraph to introduce the epidemiology of CC and the current status of treatment and prognosis. The second paragraph to introduce the importance of nutrition and assessment criteria, and the knowledge gap regarding the impact of PNI on the QoL in CC patients. The final paragraph to propose your hypothesis.
2.The rationale of the study should be emphasized. The clinical significance of this study is limited. The nutrition status is well-known to be correlated with the systemic condition, ECOG PS and QoL, therefore the prognosis. Then what’s the novelty of your study? Also, in the introduction section, this issue should be more clearly described.
3.The article structure is fine. Raw data shared. Figure 1 needs to be refined with more information.

Experimental design

This study evaluated the impact of PNI on QoL and prognosis in CC patients. Though interesting, the methodology is not sufficient and solid. In addition, how research fills an identified knowledge gap is not well described. Some relevant comments are as follows:
1. As stated in line 114, this was a prospective study. The author needs to register on clinicaltrial.com for this study and the registration No. should be provided. The study design needs to be clarified in the section of Patients or as a separate section.
2. The sample size is small. Please discuss this limitation or justify the sample size calculation.
3. Patients-selection bias may exist. The author stated “ A convenience sampling method was used”. Please clarify this convenience sampling method or provide supportive reference.
4. PSM was used to balance the baseline differences between groups. However, considering the sample size is relatively small, multivariate regression analysis should be performed to confirm the result and to avoid data loss. Also, subgroup analysis is suggested.
5. The author evaluated the difference of QoL, anti-cancer treatment response and survival between patients with low and high-PNI groups. PNI was actually affected by multifaceted factors for example the baseline ECOG PS, disease stage, which may underlyingly affect the prognosis. Hence, the interaction between these factors is suggested to be analyzed.

Validity of the findings

1. The multivariate regression analysis and subgroup analysis are recommended to validate the findings after PSM.
2. Analysis of interaction between PNI and baseline characteristics are suggested.
3. Figure 1. No. of patients in every step and that meeting inclusion/exclusion should be displayed in the Flow chart.

Reviewer 3 ·

Basic reporting

It is an interesting topic how the Prognostic Nutritional Index is used to predict prognosis and associate quality of life in women with cervical cancer.
- This reviewer is not eligible to assess writing in English
- Introduction should include the validity of using the IPN in patients with cancer in general and/or cervical cancer.

Experimental design

- The main aspect is consistency about the aim of the study. The abstract distinguishes the use of the IPN to predict patient prognosis and compare their quality of life; In the introduction (line 97) it is mentioned that it is used to predict survival quality, while in the discussion it refers to the association of IPN with the acute and late toxicity of this disease.

- Since there is a non-probabilistic sample collection, please describe the characteristics that were met by the women included in the study, and why the entire population registered in the 3 years that the data collection lasted was not included

Validity of the findings

- The QoL analysis should include information from the matching analysis data. Or display the results of the unpaired analysis as the paired one, as was the case for the survival analysis

---

## Round 0.2 · accepted · Accept

Authors have addressed all the reviewers' comments. I have assessed the revision myself and I am satisfied with the current version. The paper is ready for publication.

Reviewer 1 ·

Basic reporting

OK

Experimental design

OK

Validity of the findings

OK

Additional comments

The authors have thoroughly addressed all the edits requested. I suggest this paper is accepted with no further revisions.

Reviewer 2 ·

Basic reporting

The manuscript has been greatly improved.

Experimental design

All comments have been well addressed with supporting data or discussed as a limitation.

Validity of the findings

Additional analysis as suggested have been conducted to further validate the findings

Reviewer 3 ·

Basic reporting

Authors have responded to the comments. Accepted.

Experimental design

Authors have responded to the comments. Accepted.

Validity of the findings

Authors have responded to the comments. Accepted.

Additional comments

Authors have responded to the comments. Accepted.